# How is vocabulary involved in second language reading comprehension? A study in Chinese-English bilingual children

Qiuzhi Xie[1]*, Susanna Siu-Sze Yeung[2]*

1 Australian Centre for the Advancement of Literacy, Australian Catholic University, Sydney, Australia,
2 Department of Psychology, The Education University of Hong Kong, Hong Kong, Hong Kong

* qiuzhi.xie@acu.edu.au (QX); siusze@eduhk.hk (SSY)

## Abstract

It remains unclear as to how vocabulary contributes to reading comprehension in a second language (L2). This study aims to explore the specific roles of vocabulary in reading comprehension in children learning English as an L2 based on three theoretical perspectives. Namely, whether vocabulary should be considered as a subcomponent of language comprehension, an independent predictor of reading comprehension, or an indirect predictor of reading comprehension through decoding and listening comprehension? A total of 167 Grade 4 and 5 Chinese primary school English learners with a mean age of 9.96 years old in Hong Kong participated in this study, and they completed a series of English language tasks. The measurement models indicated that vocabulary was strongly associated with both decoding and listening comprehension, explaining above 65% of their variance. The results of structural equation modeling indicate that vocabulary substantially and indirectly contributed to reading comprehension (accounting for around 44% of its variance) through decoding and language comprehension. Theoretical and practical implications are discussed.

In today's globalized world, mastering English has become an essential ability. Learning to read in English attracts much attention in the field of education [1–3]. A widely recognized theoretical model for explaining reading comprehension is Simple View of Reading [4,5], which posits that reading comprehension is supported by two fundamental processes: decoding and language comprehension.

Several theoretical extensions of the Simple View of Reading underscore the significance of vocabulary in reading comprehension [6,7]. However, the roles of vocabulary in reading comprehension, especially among second language (L2) learners, remain unclear.

**Data availability statement:** The files including databases, materials, and analysis code are available from Open Science Framework: https://osf.io/xwda3/?view_only=0eb8ef-81d1e14ad3ac5b9d85d15631b4.

**Funding:** This study is supported by RGC Research Fund (18603717) from University Grants Committee in Hong Kong. The funder had no role in study design, data collection and analysis, decision to publish, or preparation of the manuscript.

**Competing interests:** The authors have declared that no competing interests exist.

### Is vocabulary independent of decoding and language comprehension?

A substantial body of research supports the position that language comprehension comprises the dimensions of vocabulary and listening comprehension [7,8]. However, a number of studies using regression showed that vocabulary had small significant predictive power on reading comprehension beyond decoding and listening comprehension [7,9] and, hence, this raises a debate as to whether vocabulary should be regarded as an additional independent predictor of reading comprehension along with decoding and language comprehension [10].

Researchers provided explanations for vocabulary's unique predictive power found in regression. While vocabulary and other listening comprehension instruments may indicate a single underlying construct, vocabulary instruments may be more robust to indicate this construct than other listening comprehension instruments. Alternatively, the listening comprehension instruments used in earlier studies were of lower psychometric properties than the vocabulary instruments, making them less effective at capturing the multiple aspects of language comprehension skills [10,11]. However, one problem of regression is that it does not consider the complex interactions between independent variables.

Several studies using factor analysis did not support vocabulary as a unique predictor of reading comprehension. For example, Tunmer and Chapman [7] using exploratory factor analysis found that vocabulary and listening comprehension loaded on one rather than two factors. Braze et al.'s [11] and Sabatini et al.'s [12] confirmatory factor analyses (CFA) showed that vocabulary should be structured in language comprehension rather than a unique contributor to reading comprehension. However, these studies focused on adults and on English as a first language (L1), and it remains unclear whether vocabulary has a unique contribution to reading comprehension in an L2.

### The contribution of vocabulary to decoding and language comprehension

The Reading Systems Framework [6] indicates that lexicon (including word meaning, morphology, and syntax) links the word identification and comprehension systems in reading comprehension. In addition, a recent Direct and Indirect Effects Model of Reading [13–15] indicates that vocabulary interacts with the phonological and semantic aspects in word reading while also supporting higher-order cognition and regulation (e.g., inference making and comprehension monitoring) in language comprehension. Both the two theoretical frameworks indicate that vocabulary contributes to both decoding and language comprehension. Several empirical findings supported these theoretical models by showing that vocabulary predicted reading comprehension indirectly through its contribution to both word decoding and language comprehension in primary school children [13–19].

Scholars have explained the relation between vocabulary and decoding. Walley et al. [20] argued that increasing vocabulary size strengthens phonological representation, which is crucial for word decoding. In addition, according to the Computational Models of Reading [21], phonology, semantics and orthography of words associate with each other.

The roles of vocabulary in reading comprehension may vary as a function of language proficiency. Several studies reported that vocabulary had a direct contribution to reading comprehension beyond decoding and language comprehension in older readers [22,23]; however, in beginning readers, the effect of vocabulary on reading comprehension was fully mediated through decoding and language comprehension [13,19,24].

## Vocabulary in L2 reading comprehension

Relative to their L1, learners usually have smaller vocabulary size and fewer connections between words in their L2 [19,25,26]. Several scholars studied L1 versus L2 learners and found that vocabulary was more strongly associated with decoding and reading comprehension in learners' L2 than L1. For instance, Droop and Verhoeven's [27] study in Grade 3 and 4 primary school children showed that vocabulary explained more variance in Dutch reading comprehension in L2 learners (migrant children) than in L1 learners (native Dutch children). Cho et al. [17] as well as Geva and Yaghoub [28] reported significant effect of vocabulary on decoding only in English learners but not in native English-speaking children. Pasquarella et al. [29] even found that vocabulary and decoding were distinct factors among native English speakers whereas they appeared to load on a same factor among English learners. One possible explanation for these differences is that children's L1 oral language skills are well established prior to the onset of formal reading instruction, whereas instruction for L2 oral skills and reading often occurs concurrently [17]. Thus, vocabulary is crucial to provide phonological representations for developing decoding in an L2 [19].

All the studies mentioned so far [17,27–29] involved bilingual children living in places where their L2 was the official language and predominantly used. Vocabulary may play an even more critical role for L2 learners whose L1 is rather contrasted and dominant in their social and linguistic environment. Due to little L1-L2 phonological overlap, these learners can hardly rely on their L1 to learn L2 phonological forms. Besides, the L2 exposure is limited to them. This makes vocabulary essential, especially at an early stage of L2 learning, to enable these learners to acquire the unfamiliar L2 phonological patterns. Based on this, they can then further build other language skills like decoding and listening comprehension [19].

## The present study

This study explores the specific roles of vocabulary in reading comprehension in an L2 based on three theoretical perspectives: (1) vocabulary is a sub-component of listening comprehension, (2) vocabulary is an independent component of reading comprehension, and (3) vocabulary contributes to reading comprehension through decoding and listening comprehension.

This study was conducted in Hong Kong primary school children who learned English as a second language (ESL). In Hong Kong, children begin to learn English in kindergartens. As Cantonese is the L1 and the main medium of daily communication for most Chinese children in Hong Kong, the exposure to English outside school is limited. When children are at Grade 4 and 5 of primary school, they still did not reach high proficiency of reading in English. Cantonese has a small syllable inventory of 630 (not counting tones) [30], in stark contrast to English that has over 10,000 syllables and more complex syllabic structures [31]. Acquiring English phonology is not easy for these children, and they receive the instruction of oral language and reading at a similar time. Hence, English vocabulary may be critical for these children to acquire first the unfamiliar English phonological patterns, which are essential for the development of decoding and listening comprehension [19].

In addition, Chinese and English are very different in grammar. Compared with English grammar, Chinese grammar is less complicated and has fewer rules. Some grammatical rules of English, such as tenses and plural forms, do not apply to Chinese [32]. Therefore, it is not easy for Chinese children to master English syntax, and children are more likely to depend on key words of a sentence for comprehension [19]. Considering the children's language background, we hypothesized that for Hong Kong ESL children, vocabulary influences reading comprehension primarily through its strong contributions to decoding and listening comprehension.

## Method

### Participants

A total of 167 typically developing Chinese children (90 boys) aged from 9 to 11 years old (Mean age = 9.96, *SD* = .60) from a Hong Kong primary school participated in the present study. Among these children, 87 were in Grade 4 and 80 were in Grade 5. Parental education levels of these children ranged from Primary Education to Doctorate Degree with a median of Senior Secondary Education. These children had been learning ESL for about 6 years. Cantonese was the instructional language in the school, and English was taught as a subject of around 6–8 hours every week. This study focuses on Grade 4 and 5 students as these students were still at a comparatively early stage of learning English while they have begun to develop the abilities to comprehend short passages in oral and written forms in English.

A series of language assessments were provided to the participants at several sessions in their schools. A few students were absent in some of the sessions and did not complete all the assessments. Therefore, there were different numbers of missing data for the measures administered. But all these children completed at least 3 assessments. A total of 149 students completed all the assessments.

### Measures

We used four vocabulary assessments to measure multiple aspects of vocabulary. Receptive and expressive vocabulary assessed vocabulary breadth (the number of words acquired). Word definition and synonyms tested vocabulary depth (how well words are known). Decoding was examined via word reading accuracy and fluency. The Cronbach's α values of the assessments are shown in Table 1.

### Receptive vocabulary

The Peabody Picture Vocabulary Test – 4th edition (PPVT-4) [33] was used to test English vocabulary knowledge. This test has been used many times in Hong Kong children, and good psychometric properties have been shown [34,35]. In this test, a target word was orally presented together with four pictures, and children were asked to circle one of the pictures corresponding to that word. A total of 24 items was administrated, and each correct answer deserved 1 point.

### Expressive vocabulary

We used the first 15 items in the Expressive Picture Vocabulary subtest of the Clinical Evaluations of Language Fundamentals – 5th Edition (CELF-5) [36] that has good reliability and validity [37]. Experimenter presented pictures, and children were asked to name the pictures in English. Each item was scored from 0 to 2. For several items, 1 point was given if a child's answer was partially correct (e.g., saying "award" instead of the correct answer "trophy").

**Table 1. Descriptive statistics.**

| Variables | n | Score Range | Mean | SD | Cronbach's α | Skewness |
|---|---|---|---|---|---|---|
| Receptive vocabulary | 165 | 3-22 | 12.21 | 4.46 | .77 | .08 |
| Expressive vocabulary | 167 | 0-28 | 11.69 | 6.24 | .85 | .83 |
| Vocabulary definition | 165 | 0-16 | 1.98 | 3.37 | .84 | 2.11 |
| Synonyms | 165 | 0-10 | 2.60 | 2.30 | .77 | 1.04 |
| Word reading accuracy | 161 | 0-93 | 34.45 | 21.44 | .98 | .68 |
| Word reading fluency | 155 | 0-86 | 42.58 | 17.45 | .98 | −.29 |
| Listening comprehension | 151 | 0-14 | 3.86 | 3.36 | .82 | .70 |
| Reading comprehension | 163 | 0-28 | 13.61 | 7.42 | .91 | .17 |

## Word definition

The word definition subset of the CELF-5 [36] with 12 items was used to measure the precision of word knowledge. In this test, an experimenter orally presented in English a target word and a sentence with that word (e.g., *Tease. My dad said, "Don't <u>tease</u> your brother."*), and children were asked to explain that target word in Cantonese. Each answer was scored from 0 to 2 points depending on the precision of that answer according to the marking theme of CELF-5 [36].

## Synonyms

We adapted the first 15 items from Test 17 (Reading Vocabulary) of Woodcock-Johnson III Tests of Achievement [38] that has good reliability and validity [39]. In this test, children were orally presented with a word and were asked to speak another word with the same or similar meaning (e.g., *the synonym of the target word "Lady" can be woman, female, madam, and miss*). Each correct answer was scored 1 point.

## Word reading accuracy

This task combined 2 tests used before. The first was adopted from Tong and McBride-Chang [40] which includes 60 English words. The second test contained 35 English words adopted from Test 1 (Letter-Word Identification) of Woodcock-Johnson III Test of Achievement [38]. For both the tests, the difficulty levels gradually increased.

Children began with Tong and McBride-Chang's test [40] and were asked to read the printed words one by one. Each correct answer deserved 1 point. The test stopped when a child provided five consecutively incorrect answers. Both tests were administered for each participant.

## Word reading fluency

The Sight Word Efficiency subset of the Test of Word Reading Efficiency [41] was adopted to assess children's reading fluency. Children were presented with a list of 104 English printed words with increasing difficulty and asked to read as many words as possible within 45 seconds. The number of the words that children read correctly in 45 seconds was used to indicate decoding fluency.

## Listening comprehension

The Understanding Spoken Paragraphs subset from the CELF – 4th edition [42] was used to assess children's English listening comprehension. Three stories (narrative texts) in this subtest for children aged 7–8 years old were selected. Story A was less difficult than Stories B and C. Children were orally presented with each story, followed by 5 oral questions. Three of the questions regarded the main idea, details, and sequence of the story; and the remaining 2 questions required inference (e.g., *"Why was Marcus worried?"*) and prediction making (e.g., *"Who do you think Marcus will miss seeing while he's at school on the first day?"*). A trial was provided to familiarize children with the task. Each correct answer was scored 1 point.

## Reading comprehension

This task was developed for Hong Kong primary school children by Tong and her colleagues [43] and had good psychometric properties. In this task, children were asked to read two passages (one expository and one narrative text) in English, and each passage was followed by 12 multiple-choice questions that tested their basic understanding of the passage and 3 short answer questions that required inference. Each correct answer to the multiple-choice questions was scored 1 point. The scoring of the short answer questions was in accordance with the coding scheme determined by Tong et al. [43]. The maximum score of this task was 30.

## Procedures

Ethical approval was obtained from the Education University of Hong Kong, and written consent from parents were also obtained before our data collection. The audio stimuli used for the receptive vocabulary and listening comprehension tasks were pre-recorded using a female voice. Measures involving oral responses were conducted individually by trained research assistants, while the measures of receptive vocabulary and reading comprehension were administered in groups with each group composed of around 20 children. The whole testing time was around 90 minutes for each child. The data and materials are available from Open Science Framework: https://osf.io/xwda3/?view_only=0eb8ef81d1e14ad3ac5b9d85d15631b4

## Data analyses

We firstly conducted CFA to compare the three measurement models: vocabulary as a subcomponent of language comprehension, vocabulary as an independent factor along with decoding and listening comprehension, and vocabulary as a covariate of decoding and listening comprehension. Based on the most appropriate measurement model, we then conducted structural equation modeling (SEM) to predict reading comprehension.

## Results

The descriptive statistics are displayed in Table 1. All the measures used in this study were of good internal consistency reliability as shown by the Cronbach's alpha values that were .77 or above. Vocabulary definition was positive skewed as indicated by its skewness value (2.11). The scores on the other measures were generally normally distributed, indicated by the absolute values of their skewness scores smaller than 2.00 [44].

Zero-order correlations were conducted among the language measures used in this study (Table 2). Pairwise deletion was used to deal with the missing data (the same for the partial correlations below but not for the SEM). The four vocabulary measures were significantly correlated with each other ($.42 \leq r \leq .70$, $p < .001$). Word reading accuracy and fluency were very strongly correlated ($r = .85$, $p < .001$). In addition, the vocabulary measures were significantly correlated with word reading and listening comprehension ($.45 \leq r \leq .74$, $p < .001$). Reading comprehension was significantly correlated with all the other measures ($.43 \leq r \leq .65$, $p < .001$). We additionally did partial correlations among these measures by controlling grade (Table 2) and found that the results were similar to those of zero-order correlations.

We conducted measurement models by using MPlus 8.1. Maximum likelihood estimation was used (the same below). We firstly tested a 3-factor (decoding, vocabulary, language comprehension) versus a 2-factor (decoding and language comprehension) measurement model via CFA. We only had one measurement of listening comprehension. To avoid single-indicator latent factor of language comprehension in the 3-factor model, we separated the score on Story A in the

**Table 2. Zero-order correlations (above diagonal) and partial correlations controlling for grade (below diagonal).**

|  | 1 | 2 | 3 | 4 | 5 | 6 | 7 | 8 |
|---|---|---|---|---|---|---|---|---|
| **1. Receptive vocabulary** | -- | .42 | .43 | .55 | .46 | .45 | .45 | .52 |
| **2. Expressive vocabulary** | .42 | -- | .62 | .70 | .63 | .56 | .59 | .53 |
| **3. Word definition** | .46 | .62 | -- | .65 | .58 | .45 | .54 | .43 |
| **4. Synonyms** | .57 | .70 | .65 | -- | .74 | .65 | .64 | .65 |
| **5. Word reading accuracy** | .45 | .63 | .58 | .74 | -- | .85 | .59 | .64 |
| **6. Word reading fluency** | .44 | .55 | .45 | .65 | .84 | -- | .53 | .63 |
| **7. Listening comprehension** | .43 | .59 | .54 | .64 | .58 | .52 | -- | .56 |
| **8. Reading comprehension** | .54 | .53 | .43 | .65 | .64 | .63 | .56 | -- |

All the correlations were significant ($p < .001$). Asterisks are not presented to make the table neat.

listening comprehension measure from the total score on the Stories B and C given their difficulty levels. In addition, as we had four vocabulary measurement scores and to prevent the latent language comprehension factor in the 2-factor model being predominantly represented by vocabulary, we summed up the z scores of receptive and expressive vocabulary as the vocabulary breadth composite score and we, in the same way, calculated the vocabulary depth composite score by combining the z scores of vocabulary definition and synonyms. In the 3-factor model, the latent vocabulary factor was indicated by the two vocabulary composite scores, and the latent language comprehension factor was indicated by the two listening comprehension scores (score on Story A and that on Stories B and C). In the 2-factor model, the latent language comprehension factor was indicated by the two vocabulary composite scores and the two listening comprehension scores. The correlation between the two listening comprehension scores was considered in the 2-factor model. In both the models, word reading accuracy and fluency were the two indicators of the latent decoding factor, and the correlations between latent factors were calculated.

Regarding the 3-factor CFA, the model fitted our data well: $\chi^2$ (6) = 7.321 ($p > .05$), CFI = .998, TLI = .995, RMSEA = .036 (90% CI from.000 to.111), AIC = 2524.076, BIC = 2589.554. All the indicators significantly loaded on the corresponding latent factors. However, the latent vocabulary factor was very strongly correlated with the factors of decoding ($r = .83$) and language comprehension ($r = .81$). These strong correlations did not support vocabulary as an independent factor in addition to decoding and language comprehension; rather, vocabulary seemed to be largely intertwined with decoding and listening comprehension. The correlation between decoding and language comprehension was moderately high ($r = .68$).

The 2-factor model also fitted our data well: $\chi^2$ (7) = 7.411 ($p > .05$), CFI = .999, TLI = .999, RMSEA = .019 (90% CI from .000 to .098), AIC = 2522.166, BIC = 2584.526. The two vocabulary composite scores and two listening comprehension scores all significantly loaded on the latent language comprehension factor. However, in this model, the latent decoding and language comprehension factors were found to be very highly correlated ($r = .83$). This high correlation indicates that these two factors were largely intertwined rather than independent.

We further conducted a measurement model by considering the vocabulary factor as the covariate of the decoding and language comprehension factors (Fig 1). The indicators of the latent factors were the same as the 3-factor model. The model fit was good and the same as the 3-factor model: $\chi^2$ (6) = 7.321 ($p > .05$), CFI = .998, TLI = .995, RMSEA = .036 (90% CI from .000 to .111), AIC = 2524.076, BIC = 2589.554. Vocabulary strongly contributed to both decoding (standardized $\beta = .83$) and language comprehension (standardized $\beta = .81$) by explaining around 68.1% of the variance in decoding and 65.7% of the variance in listening comprehension. After vocabulary was controlled, the correlation between decoding and language comprehension was no longer significant (See Fig 1).

The high correlations ($r > .80$) in the 3-factor or the 2-factor measurement models indicate that the multicollinearity among the factors was a serious problem [45]. Therefore, we could not predict reading comprehension based on either of the two models; otherwise, the strong multicollinearity between the predictors would probably lead to fallacious path coefficient estimates or even mistakenly lead to statistical non-significance of the parameter estimates [46]. Therefore, the last measurement model presented in Fig 1 was accepted. To see whether the results based on this model differed between Grade 4 and 5 students, we further conducted fully constrained multigroup SEM by restricting all the loadings, correlations, and paths of the model to be equal for the two grade groups. We found that $\chi^2$ (22) = 32.019 ($p > .05$). The nonsignificant $\chi^2$ (df) value indicates no significant grade difference of this model; hence, we merely presented the whole group standardized $\beta$ coefficients (see Fig 1).

We conducted the SEM to predict reading comprehension based on the last measurement model (see Fig 2). Reading comprehension was the dependent variable. The model fit was good: $\chi^2$ (10) = 15.036 ($p > .05$), CFI = .993, TLI = .986, RMSEA = .055 (90% CI from .000 to .108). In this model (Fig 2 and Table 3), decoding and listening comprehension were both significant predictors of reading comprehension (standardized $\beta = .44$ and .36 respectively, $p < .001$). Vocabulary strongly but indirectly contributed to reading comprehension (standardized $\beta = .66$, $p < .001$), explaining its variance up to about 43.5%, via decoding and listening comprehension (standardized $\beta = .37$ and .30 respectively, $p \leq .001$). A total of

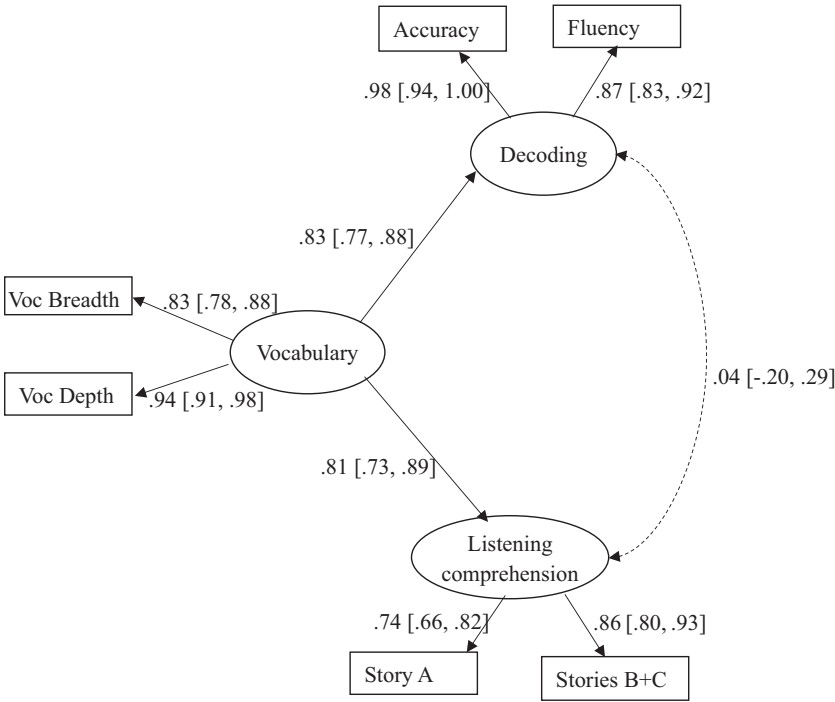

*Note.* The solid paths indicate significant effects (*p* < .001). The dashed path indicates

nonsignificant relationship. The 95% confidence interval coefficients are presented in the

brackets.

Voc = Vocabulary.

**Fig 1. Measurement model of the vocabulary, decoding, and listening comprehension factors.**

53.6% of the variance in reading comprehension was accounted for in this model. Based on this SEM, we additionally checked the direct effect of vocabulary on reading comprehension by adding a path from vocabulary to reading comprehension. Although the model fit was good: $\chi^2$ (9) = 12.441 (*p* > .05), CFI = .996, TLI = .990, RMSEA = .048 (90% CI from .000 to .106), the direct effect of vocabulary on reading comprehension was nonsignificant. We also compared the two models via the difference in $\chi^2$ with the difference in *df* and found that $\Delta\chi^2(1)$ = 2.595 (*p* > .05), suggesting that both the models fitted our data equally well; thus, the simpler model (Fig 2) was more efficient and should be recommended. To observe any grade differences based on this model (Fig 2), we conducted fully constrained multigroup SEM by restricting all the loadings, correlations, and paths for the two groups of Grade 4 and 5 students. We found that $\chi^2$ (12) = 19.821 (*p* > .05), suggesting that there was no significant grade difference; hence, only the whole group standardized $\beta$ coefficients were presented (see Table 3 and Fig 2).

## Discussion

This study advances our understanding of the roles of vocabulary in reading comprehension in those who learn an L2 at a comparatively early stage by innovatively testing the roles of vocabulary in reading comprehension based on three theoretical perspectives. The results of our measurement models did not seem to support that vocabulary was largely independent of decoding and listening comprehension. The results rather suggest that vocabulary was substantially related to

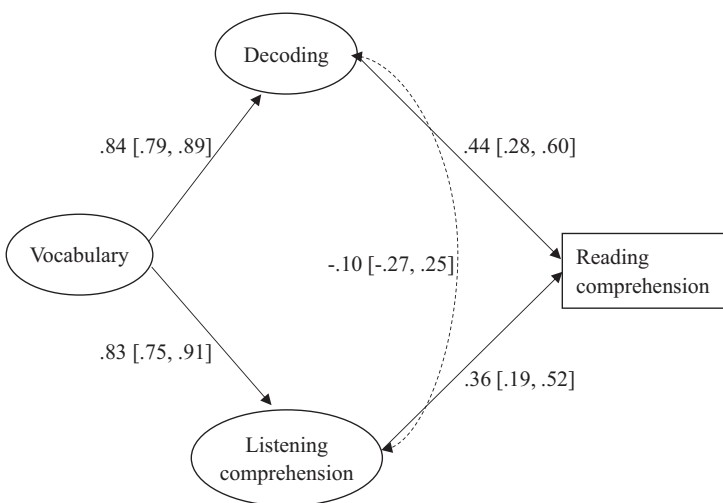

*Note.* All the solid paths indicate significant effects ($p \leq .001$). The dashed path indicates a

nonsignificant correlation. The 95% confidence interval coefficients are presented in the

brackets.

**Fig 2. Predicting reading comprehension from decoding, listening comprehension, and vocabulary.**

**Table 3. Standardized regression weights for the SEM in Fig 2.**

| Paths | | Direct effect | | | | Indirect effect | | |
|---|---|---|---|---|---|---|---|---|
| | | β | SE | p | R² | β | SE | p |
| *Factor loadings in the measurement model* | | | | | | | | |
| **Decoding** | ← Accuracy | .96 [.93,.99] | .018 | .000 | | | | |
| **Decoding** | ← Fluency | .89 [.85,.92] | .023 | .000 | | | | |
| **Listening comprehension** | ← Story A | .74 [.66,.82] | .047 | .000 | | | | |
| **Listening comprehension** | ← Stories B + C | .85 [.79,.92] | .040 | .000 | | | | |
| **Vocabulary** | ← Breadth | .83 [.78,.88] | .029 | .000 | | | | |
| **Vocabulary** | ← Depth | .94 [.90,.97] | .021 | .000 | | | | |
| *Regression weights in the structural model* | | | | | | | | |
| **Decoding** | ← Vocabulary | .84 [.79,.89] | .033 | .000 | .704 | | | |
| **Listening comprehension** | ← Vocabulary | .83 [.75,.91] | .048 | .000 | .689 | | | |
| **Reading comprehension** | | | | | .536 | | | |
| | ← Decoding | .44 [.28,.60] | .097 | .000 | | | | |
| | ← Listening comprehension | .36 [.19,.52] | .103 | .001 | | | | |
| | ← Vocabulary | | | | | .66 [.59,.74] | .045 | .000 |
| *Correlation in the structural model* | | | | | | | | |
| **Decoding ←→ Listening comprehension** | | −.01 [−.27,.25] | .159 | .950 | | | | |

Standardized coefficients were presented. The 95% confidence interval coefficients are presented in the brackets.

decoding and listening comprehension ($r > .80$ in the 3-factor model), explaining most of their variance up to above 65%. Vocabulary also accounted for the common variance between decoding and listening comprehension. The decoding and listening comprehension factors were moderately highly correlated ($r = .68$) in the 3-factor measurement model; but this correlation became nonsignificant when vocabulary was controlled as a covariate (Fig 1).

The results of our SEM show that vocabulary had a substantial indirect effect on reading comprehension (standardized $\beta = .66$, $p < .001$) via decoding and listening comprehension. These results are consistent with those in Xie et al.'s [19] longitudinal study which showed that vocabulary's contribution to reading comprehension one year later in Chinese ESL children was entirely mediated by decoding and listening comprehension. The results also echo the previous findings in beginning L1 readers [13–15,24]. However, previous studies in older readers (adolescents) showed a unique direct contribution of vocabulary to reading comprehension above decoding and listening comprehension [22,23]. It could be that the reading materials for younger reader who learn to read at an early stage are easier and less likely to involve uncommon words, and this may make the unique direct effect of vocabulary on reading comprehension less obvious in younger children [19].

Our and previous findings [13–19] support the Reading Systems Framework [6] and the Direct and Indirect Effects Model of Reading [13–15] which suggest vocabulary as critical in both word decoding and language comprehension. Vocabulary provides representations of phonological forms which are critical for decoding [20]. In addition, knowing the phonology and semantic meaning of a word can facilitate the acquisition and recognition of that word in the written form through phonology-orthography correspondence [47]. Listening comprehension also entails vocabulary knowledge. Both vocabulary breadth and depth may facilitate decoding and listening comprehension as previous studies have shown [48,49]. Having a larger size of vocabulary (breadth) leads to more knowledge of word phonological patterns and word meanings, while understanding deeper meanings (depth) of words results in better word semantic representations [50,51]. However, the roles of vocabulary may vary as a function of language proficiency. Two studies in adult L1 readers showed that vocabulary should be regarded as a language comprehension component and it was largely independent of decoding [11,12].

Compared with previous findings in an L1 and in L2 learners living in the societies where the L2 is dominant [17,18], our present findings show much stronger effects of vocabulary on decoding and listening comprehension. It could be that vocabulary knowledge plays a very critical role in both decoding and language comprehension in L2 learners who have limited L2 exposure and whose L1 is highly contrasted with the L2 [19]. Since English has a wider range of syllables and more intricate syllabic structures than Chinese [30,31], Chinese ESL learners need to first develop vocabulary that helps them grasp new phonological patterns in English. Based on the existing vocabulary knowledge, learners can further develop word decoding by matching the word phonological form with the orthographic form [19]. In addition, for these children, they receive the instruction of L2 oral language and reading skills almost simultaneously; thus, their phonological skills have not been adequately developed when they learn to read, making them more likely to depend on vocabulary as a phonological resource to develop decoding [17]. Moreover, Chinese and English are highly contrasted also in grammar [32]; hence, Chinese early-stage ESL learners who are not fluent in grammar may refer to the key words in a sentence, to a large extent, for comprehension. As a result, vocabulary may be a particularly vital resource for understanding spoken language. By contrast, L1 readers' phonological skills are usually well developed before they start to learn to read [17]. Living in an L2 context may fasten the development of phonological skills in that L2 [52]. Therefore, L1 readers and L2 readers living in an L2 context may not rely on vocabulary to a large degree to develop decoding.

This study has important practical implications for teaching and learning ESL. For early-stage ESL learners, vocabulary acquisition is crucial as it enables learners to acquire the basic phonological features of English and provides them with semantic knowledge of words that is fundamental for developing more complex language skills such as listening comprehension and decoding. Educators and parents can encourage children to listen and read more in English as language exposure is critical for vocabulary acquisition [53]. The game about producing words that have the same rhyme can enhance word phonological knowledge. In addition, using flashcards (pictures of words) can be a good way for students to broaden their vocabulary. This instructional strategy facilitates word learning via both verbal and visual perception and,

hence, creates stronger memory traces and facilitates recall [54]. Based on the words that students already have learned, educators can further emphasize the precision of word meanings as well as the specific and multiple usage of words to help students acquire vocabulary at a more in-depth level [51]. For example, using word maps for students to create diagrams around a target word with its definition, synonyms, antonyms, and collocations can enhance students' in-depth understanding of that word [55]. Creating vocabulary journals for students to record new words on a notebook with their own definitions and example sentences is another feasible instructional strategy [56].

Our findings may be limited to early-stage L2 learners with highly contrasted L1-L2 pairs. Future studies focusing on advanced L2 learners with overlapping L1-L2 phonology are needed to compare with this study. Alternatively, future comparative studies across different L1-L2 pairs and varying L2 exposure are encouraged. This will help to determine to what extent the associations observed in this study are specific to the Chinese-English sequential bilingual context or broadly applicable to L2 reading development in various contexts. These future studies will also help to show whether the roles of vocabulary in L2 reading comprehension vary as the functions of L2 proficiency and L1-L2 similarities.

This study also has methodological limitations. Listening comprehension and reading comprehension were respectively tested via one measure only and, hence, their construct validity may be limited, and the measures may underrepresent the multifaceted comprehension skills. Higher-order comprehension skills are complex and include text structure knowledge, inference making, and comprehension monitoring [57]. Therefore, future studies with multiple measures to test listening and reading comprehension are needed to enhance the construct validity and comprehensiveness of these constructs. For example, using multiple measures with different text genres (e.g., expository and narrative) and various assessment formats, such as cloze tests and summarization, to test listening and reading comprehension will more comprehensively capture the breadth of the complex higher-order comprehension skills. Furthermore, this study is cross-sectional and cannot indicate the causal relations between the variables investigated. A longitudinal design is thus necessary to investigate whether vocabulary knowledge affects the development of decoding, listening and reading comprehension.

## Conclusion

Our findings indicate that vocabulary exerted strong influence on decoding and listening comprehension, accounting for the variance they share. It also indirectly predicted reading comprehension through its impact on these two skills.

## Author contributions

**Conceptualization:** Qiuzhi Xie.

**Data curation:** Susanna Siu-Sze Yeung.

**Formal analysis:** Qiuzhi Xie.

**Funding acquisition:** Susanna Siu-Sze Yeung.

**Investigation:** Qiuzhi Xie, Susanna Siu-Sze Yeung.

**Methodology:** Qiuzhi Xie.

**Project administration:** Susanna Siu-Sze Yeung.

**Resources:** Susanna Siu-Sze Yeung.

**Supervision:** Susanna Siu-Sze Yeung.

**Validation:** Qiuzhi Xie, Susanna Siu-Sze Yeung.

**Visualization:** Qiuzhi Xie.

**Writing – original draft:** Qiuzhi Xie.

**Writing – review & editing:** Qiuzhi Xie.

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
