## [Decision Letter · Decision Letter 0]

2 Sep 2025

PONE-D-25-39563How is Vocabulary Involved in Second Language Reading Comprehension? A Study in Chinese-English Bilingual ChildrenPLOS ONE?

Dear Dr. Xie,

Thank you for submitting your manuscript to PLOS ONE. After careful consideration, we feel that it has merit but does not fully meet PLOS ONE’s publication criteria as it currently stands. Therefore, we invite you to submit a revised version of the manuscript that addresses the points raised during the review process.

Thank you for your submission. Both reviewers found the study to be methodologically sound, clearly presented, and based on robust statistical analyses. The paper contributes valuable insights into the role of vocabulary in second language reading comprehension among Chinese-English bilingual children.

The requested revisions are **minor** and focus on improving clarity and interpretive depth rather than altering the core analyses or conclusions. Specifically, please:

Streamline the literature review to reduce repetition and highlight the specific research gap.Expand the discussion of task limitations, especially the reliance on single measures for listening and reading comprehension, and emphasize the need for broader assessments in future research.Strengthen the instructional relevance by adding concrete pedagogical examples that follow from your findings.Revise figures to improve readability (larger font, clearer distinctions, more informative captions).Simplify sentence structures where possible and address the few minor language corrections noted by reviewers.

Additional recommendations, such as exploring grade-level differences, providing confidence intervals, and considering alternative approaches to missing data handling, would further enhance the paper but are not required for acceptance.

There is no conflict between the reviews—both recommend minor revision with similar suggestions for strengthening the manuscript. Once the above points are addressed, the paper will meet PLOS ONE’s criteria for publication.

We look forward to receiving your revised manuscript.

Kind regards,

Ramandeep Kaur

Academic Editor

PLOS ONE

Journal Requirements:

- https://doi.org/10.1007/s11145-022-10360-9

In your revision ensure you cite all your sources (including your own works), and quote or rephrase any duplicated text outside the methods section. Further consideration is dependent on these concerns being addressed.

4. Thank you for stating the following financial disclosure: [This study is supported by RGC Research Fund (18603717) from University Grants Committee in Hong Kong]. 

5. Please include captions for your Supporting Information files at the end of your manuscript, and update any in-text citations to match accordingly. Please see our Supporting Information guidelines for more information: http://journals.plos.org/plosone/s/supporting-information .

Additional Editor Comments:

Thank you for your submission and for presenting a carefully designed study on the role of vocabulary in L2 reading comprehension among Chinese-English bilingual children. The paper is well grounded in theory, employs robust statistical methods, and provides valuable insights into an under-researched context. Both reviewers recognize the strength and contribution of your work, and their suggestions are aimed at enhancing clarity, applicability, and interpretive depth.

Key points to address in revision:

Streamline the literature review to avoid repetition and sharpen the focus on the research gap.

Clarify construct validity limitations: as only one task each was used for listening and reading comprehension, please expand the discussion on how this affects interpretation and emphasize the importance of multiple measures in future research.

Strengthen the applied relevance by expanding the instructional implications. Consider giving concrete examples of vocabulary teaching strategies (e.g., activities targeting breadth and depth, integrating vocabulary with decoding/listening comprehension, classroom games).

Address missing data handling: while pairwise deletion was used, please add sensitivity checks (e.g., FIML or multiple imputation) or at least acknowledge the potential bias this may introduce.

Add confidence intervals alongside standardized coefficients to increase transparency.

Explore developmental patterns: given your Grade 4 and 5 participants, a brief exploratory note on potential grade-level differences (even if preliminary) would strengthen the contribution.

Revise figures for clarity (larger font, clearer line styles, more detailed captions).

Language clarity: simplify overly long sentences and correct minor errors (e.g., “there raises a debate” → “this raises a debate”; “recourses” → “resources”).

These are primarily refinements rather than fundamental changes. With these revisions, the manuscript will be well placed for acceptance.

Reviewers' comments:

Reviewer's Responses to Questions

**Comments to the Author**

1. Is the manuscript technically sound, and do the data support the conclusions?

Reviewer #1: Yes

Reviewer #2: Yes

2. Has the statistical analysis been performed appropriately and rigorously?

Reviewer #1: Yes

Reviewer #2: Yes

3. Have the authors made all data underlying the findings in their manuscript fully available?

Reviewer #1: Yes

Reviewer #2: Yes

4. Is the manuscript presented in an intelligible fashion and written in standard English?

Reviewer #1: Yes

Reviewer #2: Yes

Reviewer #1: The manuscript “How is Vocabulary Involved in Second Language Reading Comprehension? A Study in Chinese-English Bilingual Children” presents a carefully designed and theoretically motivated study that addresses an important question in second language (L2) reading research. The work is grounded in established frameworks such as the Simple View of Reading, the Direct and Indirect Effects Model of Reading, and the Reading Systems Framework. It makes a valuable contribution by testing competing hypotheses regarding the role of vocabulary in L2 reading comprehension, using robust statistical modeling techniques.

The study is technically sound, the methodology is generally rigorous, and the findings are clearly presented and supported by the data. I commend the authors for making their data and materials openly available and for clearly documenting their ethical procedures.

That said, I believe the manuscript would benefit from several areas of improvement before publication. My detailed comments are below.

- The literature review is comprehensive but somewhat repetitive. Streamlining the introduction would sharpen the focus on the specific research gap this study addresses.

- Only one task was used for listening comprehension and one for reading comprehension. This limits construct validity and may not fully capture the multidimensionality of these constructs. While the limitation is noted, the discussion should be strengthened by considering how this may affect interpretation of the results and by emphasizing the importance of multiple measures in future research.

- The discussion briefly touches on instructional relevance. Expanding this section would enhance the manuscript’s applied value. The authors could, for example, suggest vocabulary instruction strategies that target both breadth and depth, or highlight interventions that integrate vocabulary development with decoding and listening comprehension skills.

- The use of pairwise deletion is acknowledged, but this method can introduce bias. I recommend conducting sensitivity analyses with more robust approaches such as Full Information Maximum Likelihood (FIML) or multiple imputation to confirm the robustness of the findings.

- While standardized coefficients are presented, including confidence intervals would add transparency and strengthen interpretation.

- The inclusion of both Grade 4 and Grade 5 students offers an opportunity to explore developmental differences. Beyond controlling for grade, a multi-group SEM or exploratory analysis could clarify whether the pathways differ by grade level.

- The manuscript is generally well written, but several sentences are overly long and complex, which may reduce readability. Simplifying sentence structures would improve clarity. A few minor corrections are also needed, for example:

- “there raises a debate” → “this raises a debate”

- “recourses” → “resources”

Reviewer #2: This manuscript presents a well-designed and well-executed study that significantly enhances our understanding of the role of vocabulary in L2 reading comprehension among Chinese-English bilingual children. It offers valuable insights into L2 reading development in an under-researched setting.

The methodology, especially the use of SEM, and the clear presentation of findings contribute to its scientific value and demonstrate a robust analytical approach. Although some limitations exist, mainly related to the scope of measures and the cross-sectional design, the author acknowledges these and suggests promising directions for future research. The paper provides valuable theoretical insights and practical guidance for educators working with early-stage L2 learners.

The following is a list of recommendations for improvement:

- For future research, consider incorporating additional subtests or alternative measures for listening comprehension and reading comprehension to enhance the comprehensiveness and reliability of these constructs. For example, using different types of reading comprehension tasks (e.g., narrative vs. expository texts, various question formats) could provide a richer picture.

- A longitudinal study would significantly strengthen the causal inferences about the role of vocabulary in L2 reading development. Tracking the same cohort of students over time would allow for a more robust examination of developmental pathways and the dynamic interplay between vocabulary, decoding, and language comprehension.

- To address generalizability, future research could conduct comparative studies across different L1-L2 pairs and varying L2 exposure contexts. This would help to determine the extent to which the observed relationships are specific to the Chinese-English context or are more broadly applicable to L2 reading development.

- Revise Figure 1 and Figure 2 to improve visual clarity and figure presentation. This could involve increasing font sizes, ensuring clear distinctions between solid and dashed paths, and providing a more detailed explanation of the figures within the main text or in the figure captions.

- While practical implications are discussed, further elaboration on specific pedagogical strategies derived from the findings would be beneficial. For instance, providing concrete examples of language games or vocabulary acquisition activities tailored for early-stage ESL learners, beyond general suggestions, would add significant value.

With minor revisions, particularly to improve figure clarity and expand on practical implications, this manuscript would make a substantial contribution to the fields of second language acquisition and reading research.

**Do you want your identity to be public for this peer review?** For information about this choice, including consent withdrawal, please see our Privacy Policy

Reviewer #1: No

Reviewer #2: No

---

## [Author Response · Author response to Decision Letter 1]

20 Sep 2025

Responses to reviewers’ comments

We sincerely thank for the comments from the editor and both the reviewers that help us better this manuscript. In the revision, we have fully addressed the journal requirements and considered all the comments of the editor and the reviewers.

The editor’s comments

1. Thank you for your submission and for presenting a carefully designed study on the role of vocabulary in L2 reading comprehension among Chinese-English bilingual children. The paper is well grounded in theory, employs robust statistical methods, and provides valuable insights into an under-researched context. Both reviewers recognize the strength and contribution of your work, and their suggestions are aimed at enhancing clarity, applicability, and interpretive depth.

Thank you for these comments.

Key points to address in revision:

2. Streamline the literature review to avoid repetition and sharpen the focus on the research gap.

Thank you for this comment. We revised the section of Introduction to make it more concise.

3. Clarify construct validity limitations: as only one task each was used for listening and reading comprehension, please expand the discussion on how this affects interpretation and emphasize the importance of multiple measures in future research.

We admitted that listening comprehension and reading comprehension were respectively tested via one measure only and, hence, their construct validity may be limited, and the measures may underrepresent the multifaceted comprehension skills. We also stated that future studies with multiple measures to test listening and reading comprehension are needed to enhance the construct validity and comprehensiveness of these constructs. For example, using multiple measures with different text genres (e.g., expository and narrative) and various assessment formats, such as cloze tests and summarization, to test listening and reading comprehension will more comprehensively capture the breadth of the complex higher-order comprehension skills. Please see the last paragraph of Discussion.

4. Strengthen the applied relevance by expanding the instructional implications. Consider giving concrete examples of vocabulary teaching strategies (e.g., activities targeting breadth and depth, integrating vocabulary with decoding/listening comprehension, classroom games).

Thank you for this comment. We provided specific instructional strategies for promoting vocabulary breadth, including encourage children to listen and read more in English, the game about producing words that have the same rhyme, and using flashcards. We also suggested the strategies for promoting vocabulary depth, including word maps and creating vocabulary journals. Please refer to the last paragraph on page 21 to the 1st paragraph on page 22.

5. Address missing data handling: while pairwise deletion was used, please add sensitivity checks (e.g., FIML or multiple imputation) or at least acknowledge the potential bias this may introduce.

Pairwise deletion was only used for the correlations. We also did listwise deletion for the correlations, and the results were almost the same. For all the SEM, Maximum Likelihood estimation was used to deal with the missing data. Please see the 1st paragraph on page 14.

6. Add confidence intervals alongside standardized coefficients to increase transparency.

We provided the 95% confidence intervals on Fig 1, 2 and in Table 3.

7. Explore developmental patterns: given your Grade 4 and 5 participants, a brief exploratory note on potential grade-level differences (even if preliminary) would strengthen the contribution.

We followed Reviewer 1’s suggestion and conducted fully constrained multigroup SEM to see potential grade differences based on the accepted SEM (Fig 1 and 2). However, the values of ꭓ2 (df) were nonsignificant, indicating that there was no significant grade difference of these models. Hence, we merely presented the whole group standardized � coefficients for the SEM. Please see the 2nd paragraph on page 16 and the paragraph on page 17.

8. Revise figures for clarity (larger font, clearer line styles, more detailed captions).

To enhance the visual clarity of the figure presentation, we increased the font sizes to 12. Solid and dashed paths were used to represent significant and nonsignificant paths respectively. We changed the Figure 2 Captions to “Predicting Reading Comprehension from Decoding, Listening Comprehension, and Vocabulary” to make the explanation clearer.

9. Language clarity: simplify overly long sentences and correct minor errors (e.g., “there raises a debate” → “this raises a debate”; “recourses” → “resources”).

We made some sentences more concise and corrected these errors (e.g., changed “there raises a debate” to “this raises a debate” and changed “recourses” to “resources”).

10. These are primarily refinements rather than fundamental changes. With these revisions, the manuscript will be well placed for acceptance.

Thank you for this comment.

Reviewer #1: The manuscript “How is Vocabulary Involved in Second Language Reading Comprehension? A Study in Chinese-English Bilingual Children” presents a carefully designed and theoretically motivated study that addresses an important question in second language (L2) reading research. The work is grounded in established frameworks such as the Simple View of Reading, the Direct and Indirect Effects Model of Reading, and the Reading Systems Framework. It makes a valuable contribution by testing competing hypotheses regarding the role of vocabulary in L2 reading comprehension, using robust statistical modeling techniques.

11. The study is technically sound, the methodology is generally rigorous, and the findings are clearly presented and supported by the data. I commend the authors for making their data and materials openly available and for clearly documenting their ethical procedures.

Thank you for these comments. The data and materials are available from Open Science Framework: https://osf.io/xwda3/?view_only=0eb8ef81d1e14ad3ac5b9d85d15631b4

In addition, we provided more details about the ethical procedures that “ethical approval was obtained from the Education University of Hong Kong, and written consent from parents were also obtained before our data collection”. Please refer to the section of Procedure. The document of ethical approval is attached as a supporting document in the submission.

12.That said, I believe the manuscript would benefit from several areas of improvement before publication. My detailed comments are below.

- The literature review is comprehensive but somewhat repetitive. Streamlining the introduction would sharpen the focus on the specific research gap this study addresses.

Thank you for this comment. We revised the section of Introduction to make it more concise.

13. Only one task was used for listening comprehension and one for reading comprehension. This limits construct validity and may not fully capture the multidimensionality of these constructs. While the limitation is noted, the discussion should be strengthened by considering how this may affect interpretation of the results and by emphasizing the importance of multiple measures in future research.

We admitted that listening comprehension and reading comprehension were respectively tested via one measure only and, hence, their construct validity may be limited, and the measures may underrepresent the multifaceted comprehension skills. We also stated that future studies with multiple measures to test listening and reading comprehension are needed to enhance the construct validity and comprehensiveness of these constructs. For example, using multiple measures with different text genres (e.g., expository and narrative) and various assessment formats, such as cloze tests and summarization, to test listening and reading comprehension will more comprehensively capture the breadth of the complex higher-order comprehension skills. Please see the last paragraph of Discussion.

14. The discussion briefly touches on instructional relevance. Expanding this section would enhance the manuscript’s applied value. The authors could, for example, suggest vocabulary instruction strategies that target both breadth and depth, or highlight interventions that integrate vocabulary development with decoding and listening comprehension skills.

Thank you for this comment. We provided specific instructional strategies for promoting vocabulary breadth, including encourage children to listen and read more in English, the game about producing words that have the same rhyme, and using flashcards. We also suggested the strategies for promoting vocabulary depth, including word maps and creating vocabulary journals. Please refer to the last paragraph on page 21 to the 1st paragraph on page 22.

15. The use of pairwise deletion is acknowledged, but this method can introduce bias. I recommend conducting sensitivity analyses with more robust approaches such as Full Information Maximum Likelihood (FIML) or multiple imputation to confirm the robustness of the findings.

Pairwise deletion was only used for the correlations. We also did listwise deletion for the correlations, and the results were almost the same. For all the SEM, Maximum Likelihood estimation was used to deal with the missing data. Please see the 1st paragraph on page 14.

16. While standardized coefficients are presented, including confidence intervals would add transparency and strengthen interpretation.

We provided the 95% confidence intervals on Figure 1, 2 and in Table 3.

17. The inclusion of both Grade 4 and Grade 5 students offers an opportunity to explore developmental differences. Beyond controlling for grade, a multi-group SEM or exploratory analysis could clarify whether the pathways differ by grade level.

Thank you for this comment. We conducted fully constrained multigroup SEM to see potential grade differences based on the accepted models (Figure 1 and 2). However, the values of ꭓ2 (df) were nonsignificant, indicating that there was no significant grade difference of these models. Hence, we merely presented the whole group standardized � coefficients. Please see the 2nd paragraph on page 16 and the paragraph on page 17.

18. The manuscript is generally well written, but several sentences are overly long and complex, which may reduce readability. Simplifying sentence structures would improve clarity. A few minor corrections are also needed, for example:

- “there raises a debate” → “this raises a debate”

- “recourses” → “resources”

Thank you for these comments. We made some sentences more concise and corrected these errors (e.g., changed “there raises a debate” to “this raises a debate” and changed “recourses” to “resources”).

Reviewer #2: This manuscript presents a well-designed and well-executed study that significantly enhances our understanding of the role of vocabulary in L2 reading comprehension among Chinese-English bilingual children. It offers valuable insights into L2 reading development in an under-researched setting.

19. The methodology, especially the use of SEM, and the clear presentation of findings contribute to its scientific value and demonstrate a robust analytical approach. Although some limitations exist, mainly related to the scope of measures and the cross-sectional design, the author acknowledges these and suggests promising directions for future research. The paper provides valuable theoretical insights and practical guidance for educators working with early-stage L2 learners.

Thank you for these comments.

20. The following is a list of recommendations for improvement:

- For future research, consider incorporating additional subtests or alternative measures for listening comprehension and reading comprehension to enhance the comprehensiveness and reliability of these constructs. For example, using different types of reading comprehension tasks (e.g., narrative vs. expository texts, various question formats) could provide a richer picture.

Thank you for this comment. We admitted that listening comprehension and reading comprehension were respectively tested via one measure only and, hence, their construct validity may be limited, and the measures may underrepresent the multifaceted comprehension skills. We also stated that future studies with multiple measures to test listening and reading comprehension are needed to enhance the construct validity and comprehensiveness of these constructs. For example, using multiple measures with different text genres (e.g., expository and narrative) and various assessment formats, such as cloze tests and summarization, to test listening and reading comprehension will more comprehensively capture the breadth of the complex higher-order comprehension skills. Please see the last paragraph of Discussion.

21. A longitudinal study would significantly strengthen the causal inferences about the role of vocabulary in L2 reading development. Tracking the same cohort of students over time would allow for a more robust examination of developmental pathways and the dynamic interplay between vocabulary, decoding, and language comprehension.

Thank you for this suggestion. We stated that a longitudinal design is necessary to investigate whether vocabulary knowledge affects the development of decoding and language comprehension. Please see the last paragraph of Discussion.

22. To address generalizability, future research could conduct comparative studies across different L1-L2 pairs and varying L2 exposure contexts. This would help to determine the extent to which the observed relationships are specific to the Chinese-English context or are more broadly applicable to L2 reading development.

Thank you for this comment. We adopted this suggestion and wrote that “Future studies focusing on advanced L2 learners with overlapping L1-L2 phonology are needed to compare with this study. Alternatively, future comparative studies across different L1-L2 pairs and varying L2 exposure are encouraged. This will help to determine to what extent the associations observed in this study are specific to the Chinese-English sequential bilingual context or broadly applicable to L2 reading development in various contexts. These future studies will also help to show whether the roles of vocabulary in L2 reading comprehension vary as the functions of L2 proficiency and L1-L2 similarities.” Please see the 2nd paragraph on page 22.

23. Revise Figure 1 and Figure 2 to improve visual clarity and figure presentation. This could involve increasing font sizes, ensuring clear distinctions between solid and dashed paths, and providing a more detailed explanation of the figures within the main text or in the figure captions.

To enhance the visual clarity of the figure presentation, we increased the font sizes to 12. Solid and dashed paths were used to represent significant and nonsignificant paths respectively. We changed the Figure 2 Captions to “Predicting Reading Comprehension from Decoding, Listening Comprehension, and Vocabulary” to make the explanation clearer.

24. While practical implications are discussed, further elaboration on specific pedagogical strategies derived from the findings would be beneficial. For instance, providing concrete examples of language games or vocabulary acquisition activities tailored for early-stage ESL learners, beyond general suggestions, would add significant value.

Thank you for this comment. We provided specific instructional strategies for promoting vocabulary learning, including encourage children to listen and read more in English, the game about producing words that have the same rhyme, using flashcards, word maps, and creating vocabulary journals. All these instructional strategies are applicable for early-stage and middle-stage learners. Please refer to the last paragraph on page 21 to the 1st paragraph on page 22.

25. With minor revisions, particularly to improve figure clarity and expand on practical implications, this manuscript would make a substantial contribution to the fields of second language acquisition and reading research.

Thank you for this comment. We did all the minor revisions accordingly.

---

## [Editor Report · Decision Letter 1]

30 Sep 2025

How is Vocabulary Involved in Second Language Reading Comprehension? A Study in Chinese-English Bilingual Children

PONE-D-25-39563R1

Dear Dr. Xie,

We’re pleased to inform you that your manuscript has been judged scientifically suitable for publication and will be formally accepted for publication once it meets all outstanding technical requirements.

Kind regards,

Ramandeep Kaur

Academic Editor

PLOS ONE
---

## [Editor Report · Acceptance letter]

PONE-D-25-39563R1

PLOS ONE

Dear Dr. Xie,

I'm pleased to inform you that your manuscript has been deemed suitable for publication in PLOS ONE. Congratulations! Your manuscript is now being handed over to our production team.

Kind regards,

on behalf of

Dr. Ramandeep Kaur

Academic Editor

PLOS ONE